# The docking of synaptic vesicles on the presynaptic membrane induced by α-synuclein is modulated by lipid composition

Wing K. Man[1], Bogachan Tahirbegi[2], Michail D. Vrettas [3], Swapan Preet[1], Liming Ying [4], Michele Vendruscolo [1], Alfonso De Simone [3,5✉] & Giuliana Fusco[1✉]

α-Synuclein (αS) is a presynaptic disordered protein whose aberrant aggregation is associated with Parkinson's disease. The functional role of αS is still debated, although it has been involved in the regulation of neurotransmitter release via the interaction with synaptic vesicles (SVs). We report here a detailed characterisation of the conformational properties of αS bound to the inner and outer leaflets of the presynaptic plasma membrane (PM), using small unilamellar vesicles. Our results suggest that αS preferentially binds the inner PM leaflet. On the basis of these studies we characterise in vitro a mechanism by which αS stabilises, in a concentration-dependent manner, the docking of SVs on the PM by establishing a dynamic link between the two membranes. The study then provides evidence that changes in the lipid composition of the PM, typically associated with neurodegenerative diseases, alter the modes of binding of αS, specifically in a segment of the sequence overlapping with the non-amyloid component region. Taken together, these results reveal how lipid composition modulates the interaction of αS with the PM and underlie its functional and pathological behaviours in vitro.

[1] Centre for Misfolding Diseases, Department of Chemistry, University of Cambridge, Cambridge CB2 1EW, UK. [2] Department of Chemistry, Imperial College London, Molecular Sciences Research Hub, White City Campus, London W12 0BZ, UK. [3] Department of Pharmacy, University of Naples "Federico II", Naples 80131, Italy. [4] National Heart and Lung Institute, Imperial College London, Molecular Sciences Research Hub, White City Campus, London W12 0BZ, UK. [5] Department of Life Sciences, Imperial College London, South Kensington SW7 2AZ, UK. ✉email: alfonso.desimone@unina.it; gf203@cam.ac.uk

α-Synuclein (αS) is a 14 kDa protein whose aggregation is strongly linked with Parkinson's disease (PD) and other neurodegenerative disorders collectively known as synucleinopathies, which include dementia with Lewy bodies and multiple system atrophy[1–4]. Aggregates of αS are major components of intraneuronal inclusions known as Lewy bodies[5]. Genetic links also exist between αS and familial forms of early onset PD, including mutations, duplications and triplications of the αS encoding gene[3].

Although the function of αS is still debated[6], its prevalence at the presynaptic termini indicates that it may be involved in synaptic plasticity[7] and learning[8]. More specifically, a large body of evidence exists about a role of αS in the regulation of the homoeostasis of synaptic vesicles (SVs) during neurotransmitter release[9–14], including contexts requiring intense neuronal activity[15]. αS shows binding affinity for SVs in vitro and strongly colocalises with SVs in synaptosomes in the presence of calcium ions[16]. Upon the binding of SVs, αS has a tendency to promote their clustering[12–14,17], a process that has been associated with the maintenance of SV pools at the synaptic termini[11,18–20]. Additionally, αS has been shown to influence the regulation of the vesicle trafficking from the endoplasmic reticulum (ER) to the Golgi[13,20], and to localise at mitochondrial membranes, where it has been proposed to mitigate the effects of oxidative stress[21–24]. All these putative functions by αS require its binding to biological membranes[25,26], a central interaction that defines the relevant biological form of αS in vivo[27] and influences the kinetics of its aggregation[11,28,29], as well as the toxicity of its aggregates[1,30,31].

The characterisation of the binding mechanism of αS with synaptic membranes is therefore crucial to clarify its biological properties under physiological and pathological conditions. The intrinsic structural disorder of αS in the cytosol[32], however, is partially retained in its membrane-bound state, making it challenging to study the mechanism of membrane binding as well as the conformational properties of its membrane-associated forms. When bound with lipid bilayers, αS becomes enriched in amphipathic α-helical structure, a conformational feature promoted by seven imperfect sequence repeats in the region spanning residues 1–90[33–37]. These modular sequences provide αS the plasticity to bind a large variety of lipid membranes via multiple binding modes[35] and adopting different structural topologies such as broken[36,38] and fully extended α-helices[27,39].

We here studied the conformational properties of αS upon binding with the inner (cytosolic) and outer (extracellular) leaflets of the presynaptic membrane (PM). The results indicate a considerable preference to bind the inner PM leaflet (IPM), where αS induces in a concentration-dependent manner a stabilisation of the docking of SVs. The underlying 'double-anchor' mechanism is promoted by distinctive structural and topological properties of αS at the surface of IPM. The data then show that changes in the lipid composition of the PM, of the type occurring in neurodegenerative conditions, alter the binding and conformational properties of αS, specifically in a segment of the sequence that overlaps substantially with the non-amyloid component (NAC) region.

Taken together these results reveal how the conformational properties of αS at the inner and outer leaflets of PM are strongly influenced by the composition of the lipid membrane, and regulate its behaviour under physiological and pathological conditions.

## Results

**Structural properties of αS at the inner and outer leaflets of the presynaptic membrane.** We investigated the binding of N-terminally acetylated αS, the most common form of the protein

**Table 1 Molar fractions of the lipid bilayers used in this work.**

|  | IPM | OPM | IPM-GMs | OPM-GMs |
|---|---|---|---|---|
| PC | 0.14 | 0.253 | 0.112 | 0.2 |
| PE | 0.219 | 0.115 | 0.176 | 0.091 |
| PS | 0.101 | – | 0.080 | – |
| PI | 0.052 | – | 0.041 | – |
| PIPS | 0.015 | – | 0.012 | – |
| Cholesterol | 0.458 | 0.464 | 0.367 | 0.368 |
| Sphingomyelin | 0.016 | 0.062 | 0.012 | 0.049 |
| Cerebrosides | – | 0.077 | – | 0.061 |
| GM1 | – | 0.015 | 0.100 | 0.115 |
| GM3 | – | 0.015 | 0.100 | 0.115 |

in vivo[40], with lipid bilayers that mimic the composition of the inner (IPM) and the outer (OPM) leaflets of PM. These two leaflets share common lipid components, including phosphatidylcholine (PC), phosphatidylethanolamines (PE), sphingomyelin and cholesterol (Table 1), while featuring distinctive ones, such as phospho-L-serine (PS), phosphatidylinositol (PI) and phosphatidylinositol phosphates (PIPs), which are specific to IPM, and gangliosides (GMs) and cerebrosides, which are specific to OPM[41,42].

IPM and OPM lipid mixtures were prepared in the form of small unilamellar vesicles (SUVs) with size distributions centred around 50 nm, as measured using dynamic light scattering (DLS) (see Methods). The utilisation of this form of lipid assemblies enables a direct comparison of the present results with interaction studies previously performed with synaptic-like SUVs (SL-SUVs) composed of dioleoyl-phosphoethanolamine (DOPE), dioleoyl-phosphatidylcholine (DOPC) and dioleoyl-phosphatidylserine (DOPS)[33] and cholesterol[43].

We first examined the conformational basis of the interaction of αS with IPM and OPM using chemical exchange saturation transfer (CEST) experiments in solution nuclear magnetic resonance (NMR) spectroscopy (Fig. 1 a–c)[33,44–47]. CEST has been shown to be an accurate probe of the equilibrium between membrane-unbound and membrane-bound states of αS[17,33] that are, respectively, detectable and undetectable in solution NMR experiments. CEST is based on the selective saturation of NMR-undetectable membrane-bound states using a continuous weak radiofrequency field at offsets that range up to ±28 kHz[16,17,33]. The saturation is then observed as an attenuation of the signal intensities in the NMR-detectable unbound state of the protein as a result of the exchange between bound and unbound conformations. In probing the interaction between αS and lipid vesicles at a residue-specific resolution, the current implementation of CEST offers advantages over methods based on the signal attenuation in $^1$H-$^{15}$N-HSQC spectra. By saturating the protein resonances directly in the membrane-bound state, CEST probes exclusively the binding strength between individual residues of αS and the membranes, resulting in high sensitivity also at low lipid/protein ratios, conditions under which protein or lipid aggregation can be minimised. In addition, CEST data are largely independent from side factors that may affect transverse relaxation of the protein during the experiment, such as for example conformational exchange in the millisecond timescale.

$^1$H-$^{15}$N CEST measurements employing two different saturation frequencies (350 Hz in Figs. 1a, b and S1 and 170 Hz in Fig. S2) were carried out on a sample composed of αS (300 μM) and IPM or OPM SUVs (0.6 mg/ml). The measured CEST profiles revealed a significant difference in the binding of αS with the two leaflets of the PM. In particular, the protein showed moderate binding affinity for IPM, with high saturation primarily found in

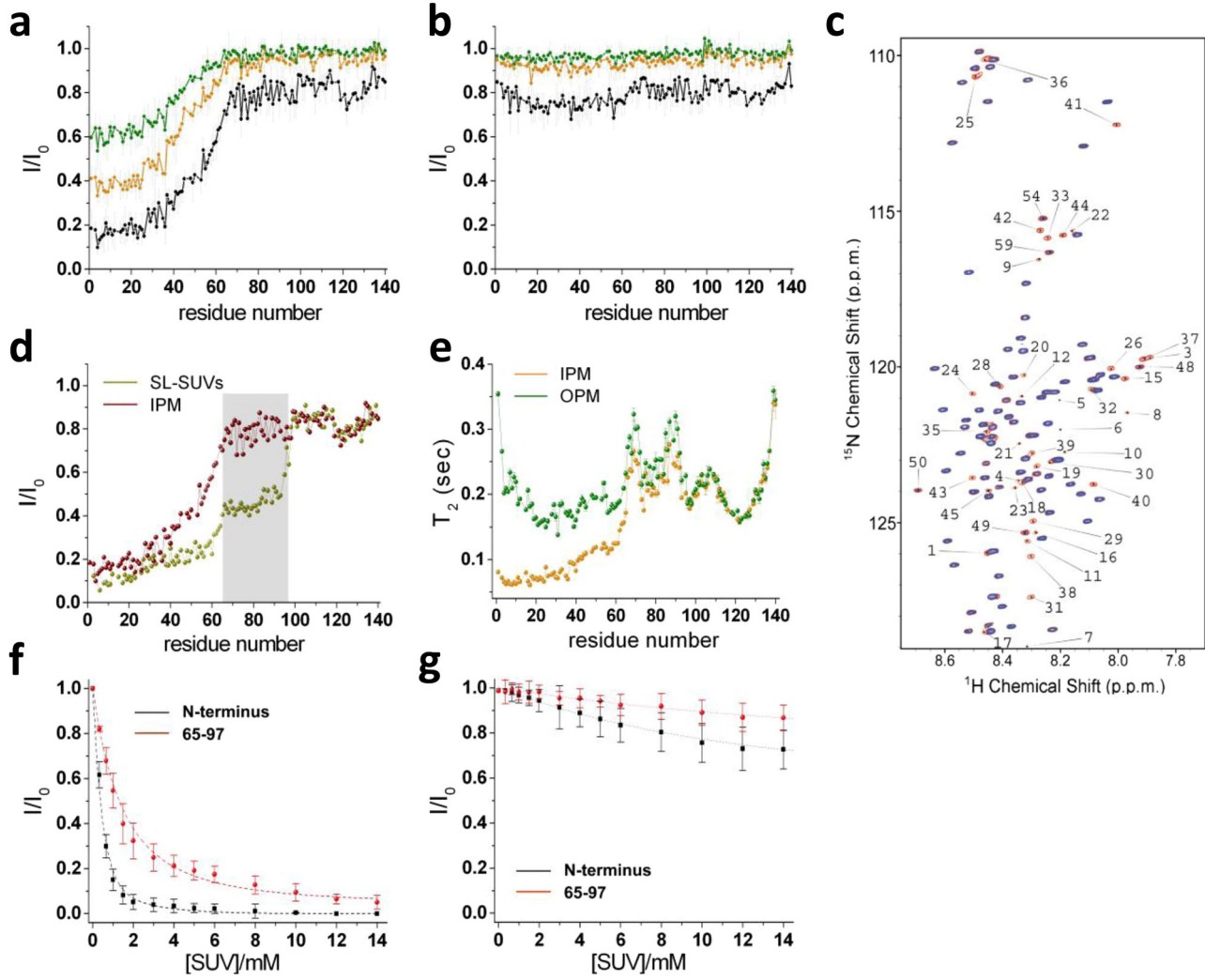

**Fig. 1 αS binds IPM more strongly than OPM.** Interaction between αS and IPM (**a**) and OPM (**b**) monitored using NMR CEST. Spectra were aquired at 283 K in 20 mM of phosphate buffer at pH 6.0, at a [1]H frequency of 700 MHz, using a protein and SUV IPM concentrations of 300 μM and 0.06% (0.6 mg/ml), respectively. NMR CEST profiles measured using a saturation bandwidth of 350 Hz (Fig. S2 for measurements performed with a bandwidth of 170 Hz) and probing the interaction between αS and IPM (**a**) or OPM (**b**). Black, orange and green lines refer to the averaged CEST profiles measured using offsets at ±1.5, ±3.0, and ±5.0 kHz, respectively. Error bars report the standard deviation estimated on the triplicate measurements. **c** Representative [1]H-[15]N-HSQC CEST spectra of αS in the presence of IPM measured using a 350 Hz continuous wavelength at offsets of 100 kHz (red) and 1.5 kHz (blue). **d** Comparison of the interaction of αS with SL-SUVs[43] (yellow) and IPM (red) probed with NMR CEST profiles measured using a saturation bandwidth of 350 Hz and offsets of ±1.5 kHz. Data for the SL-SUVs binding[43] were measured at protein and lipid concentrations of 300 μM and 0.06% (0.6 mg/ml), respectively. Grey background highlights the significant difference in the saturation of the region spanning residues 65–97, resulting in populations of detached conformations for this region of 51% and 95% for SL-SUVs and IPM, respectively. **e** $T_2$ values from transverse relaxation measurements (experimental conditions as in **a**, **b**). Green and orange report $T_2$ values of αS in the presence of OPM and IPM, respectively. Error bars report the $T_2$ fitting error. **f–g** Binding curves of αS to SUVs monitored via the signal attenuation of the peaks in the [1]H-[15]N-HSQC spectra of αS (50 μM) as a function of the concentration of SUVs. The signal attenuations have been averaged across the residues of the N-terminal region (black) and the region spanning residues 65–97 (red) and error bars report the standard deviation of these values. In the case of IPM, the fitting provided $K_D$ values of 5.2 μM ($L = 14.3$) and 88.9 μM ($L = 10.3$) for the N-terminal (residues 1–25) and central (residues 65–97) region, respectively, whereas for OPM $K_D$ values resulted respectively 5933 μM ($L = 5.7$) and 13,689 μM ($L = 6.1$).

the acetylated N terminus of αS (Fig. 1a). The binding strength for IPM SUVs of this region resulted to be similar to that observed in the case of SL-SUVs (Fig. 1d), where the N terminus has a role of anchor for the membrane binding by αS[33,48]. The saturation effects were found to decrease gradually in the residues following the N-terminal anchor, with a sharp transition observed in the region spanning residues 40–64 (Fig. 1a) leading to an almost complete loss of membrane affinity in the region 65–140. This finding therefore reveals that αS adopts unusual conformational properties when bound to the surface of IPM. In particular the region 65–97, which overlaps in large part with the amyloidogenic

NAC region[5,49], resulted to be mostly unbound to IPM surfaces up to the same level of the negatively charged region 98–140, which generally shows poor affinity for lipid membranes[33,35] and detergent micelles[38].

A different scenario was observed in the study of the αS interaction with OPM. In particular, CEST measurements resulted in very similar saturation profiles to those of the isolated αS in solution[33] (Figs. 1b, S1 and S2). When plotting the CEST profiles along the sequence, only the experiments performed using offsets at ±1.5 kHz and a saturation frequency of 350 Hz showed some very mild saturation levels in the region

of residues 1–65 of the protein, but this effect resulted too weak to be detected at other offsets or using a saturation frequency of 170 Hz.

Further details on the binding of αS with IPM and OPM were obtained using experiments of [15]N transverse relaxation (Fig. 1e). In the case of IPM, these measurements reported low $T_2$ values for the N-terminal region, indicating strong local membrane interactions, and high values of $T_2$ in the regions 65–97 and 98–140, indicating negligible membrane binding. Conversely, in the case of OPM, [15]N transverse relaxation experiments showed high values of $T_2$ throughout the membrane-binding region.

In order to quantify the binding affinity for IPM and OPM, we analyzed the signal attenuation in [1]H-[15]N-HSQC spectra of αS (50 μM) in the presence of increasing amounts of SUVs (from 0 to 14 mM), and averaged these data across two independent protein regions having distinct roles in the interaction with biological membranes, namely the N-terminal anchor (residues 1–25) and the region spanning residues 65–97. The resulting binding curves were fitted using a quadratic expression[50] (Eq. 1) that provides both the dissociation constant, $K_D$, and the number of lipid molecules, $L$, interacting with a single αS molecule:

$$\chi_B = \frac{\left([\alpha S] + \frac{[SUVs]}{L} + K_D\right) - \sqrt{\left([\alpha S] + \frac{[SUVs]}{L} + K_D\right)^2 - \frac{4[SUVs][\alpha S]}{L}}}{2[\alpha S]}$$

(1)

where $\chi_B$ is the fraction of bound protein estimated as $(1 - \frac{I}{I_0})$ for each residue and averaged across the analyzed protein region. The results indicated that the N terminus of αS interacts strongly with IPM ($K_D = 5.2$ μM and $L = 14.3$) whereas the region 65–97 has a nearly 18-fold decrease in affinity for these membranes ($K_D = 88.9$ μM and $L = 10.3$) (Fig. 1f). By contrast, αS binding with OPM (Fig. 1g) was found to be very weak for both the N terminus ($K_D = 5933$ μM and $L = 5.7$) and for the region 65–97 ($K_D = 13{,}689$ μM and $L = 6.1$). A significant difference in the interaction with IPM and OPM was also detected using circular dichroism (CD), which probes the conformational changes from disordered-unbound to helical membrane-bound states of αS (Fig. S3).

**Mechanism by which αS assists SV docking to IPM.** The study of the binding modes of αS with IPM indicated a very weak membrane affinity in the region spanning residues 65–97 (Fig. 1f) when compared with the interaction with SL-SUVs (Fig. 1d). In particular, conformations featuring a membrane-unbound region 65–97 were found to be 95% in the case of IPM and 51% in the case of SL-SUVs[43]. This significant difference is expected to result in different behaviours adopted by αS at the surface of these two membrane types, as indeed the membrane affinity of the region 65–97 is a key determinant of its biological properties[17,33]. This region is indeed a crucial driver of the affinity of αS for acidic membranes[33] and plays a role in the promotion of SV clustering via a 'double-anchor' mechanism[17] in which the N-terminal region (first anchor) and the region 65–97 (second anchor) bind simultaneously two different vesicles up to a distance of 150 Å[17].

The analysis of the conformational properties at the surface of IPM indicates that, upon binding via the N-terminal anchor, the region spanning residues 65–97 remains largely available to engage in interactions with other molecules and organelles, including SVs. This finding suggests that the conformations adopted by the IPM-bound αS are particularly active to stabilise the docking of SV on the IPM surface (Fig. 2a). The stronger affinity of the region 65–97 to bind the membrane component of SVs as respect to IPM (Fig. 1d) also indicates that this protein segment would preferentially interact with SVs over IPM, thereby

suggesting a topological preference in the double-anchor mechanism.

To generate quantitative analysis of the stabilisation of SV docking onto the PM by αS, we used total internal reflection fluorescence (TIRF) microscopy, which enabled to sample the nature of hundreds of docking events in individual experiments (Fig. 2b, c). TIRF imaging was carried out by fluorescently labelling SL-SUVs using 0.1% Topfluor® PC lipids and by forming an IPM lipid bilayer on glass surfaces via overnight incubation at 4 °C of IPM SUVs (Methods). As the focal plane of our TIRF setup extends up to 150 nm from the glass surface (Fig. 2c), the resulting images capture SL-SUVs that are in focus when they dock or are in close proximity of the IPM lipid bilayer. Vesicles in the bulk solution remain out of focus and their blurred bright dot images can be filtered out during the image processing (see Methods).

TIRF experiments were performed by incubating fluorescently labelled SL-SUVs in glass wells coated with IPM lipid bilayers. TIRF image sequences were recorded for 300 s with a frame rate of 25 frames per second (Methods) after a photobleaching step (160 s) that removed the contribution of vesicles docked prior to the sampling. Measurements were first performed in the absence of αS and subsequently by adding increasing concentrations of the protein in the same well. The results show that by maintaining a constant concentration of SL-SUVs in the well (2 μM) the number of vesicles docking onto the IPM surface increases in the presence of increasing concentrations of αS (Fig. 2d). In particular, under these experimental conditions, the strongest increase in SL-SUVs docking was observed in measurements made using 10 μM αS, with $26.8 \pm 2.1$ vesicles sampled per frame compared to $11.5 \pm 2.3$ observed in the absence of αS. An additional increase to $37.6 \pm 6.4$ vesicles in the images was measured when αS concentration was 100 μM. In addition to modifying the propensity of docking, our experiments indicated that αS alters the time spent by SL-SUVs onto IPM surfaces (Fig. 2e). In particular, by calculating the autocorrelation function (ACF) of the vesicles in the TIRF focal plane (Fig. S4), we estimated the residence time of docked SL-SUVs increasing from $440 \pm 8$ to $900 \pm 9$ ms in the presence of 0 and 10 μM αS, respectively. An additional 10% increase of the residence time was found when the concentration of αS was set to 100 μM. The observed concentration dependence suggests that multiple αS molecules can simultaneously contribute to the stabilisation of the docking of a single vesicle.

**Changes in the PM composition modulate the properties of αS at the surface of IPM and OPM.** In some neurodegenerative conditions, increased amounts of GMs in neuronal membranes have been observed[51]. Alterations in the GM fractions were found in analyses of Alzheimer's disease brains, resulting in an increase of GM clustering at presynaptic neuritic terminals[52]. Analyses of brains from Alzheimer's patients also found high concentration of GMs in detergent-resistant membrane fractions from the frontal cortex and the temporal cortex[53] as well as in lipid rafts[54]. In addition, age-dependent mechanisms were also shown to induce increase in the density of GMs in some neuronal populations[52]. An increase in the GM content of the cellular membrane is also associated with enhanced toxicity of misfolded protein oligomers[55]. αS has significant propensity for in vitro binding of SUVs containing monosialotetrahexosylganglioside (GM1)[56,57] as well as liquid-ordered lipid domains of cytoplasmic membranes that are rich in GM1[58]. It has also been proposed that elevated concentrations of GM1 in lipid rafts favour the cell internalisation of αS[59]. The effect of GMs on the aggregation properties of αS are currently debated, with both inhibition[56] or

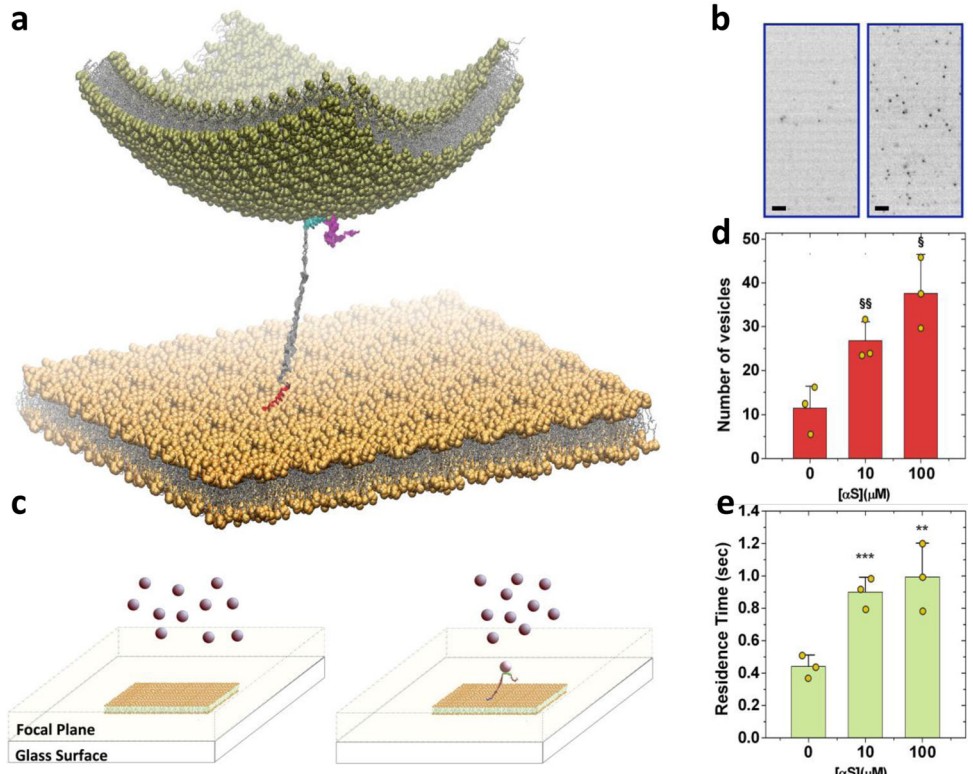

**Fig. 2 αS mediates the docking of SVs to IPM. a** Illustration of the mechanism by which αS tethers the outer leaflet of a SV (green vesicle) to the inner leaflet of a synaptic membrane (flat yellow membrane). A double-anchor conformation of αS was modelled with the N-terminal region (red) bound to the IPM and the region of residues 65–97 (cyan) tethering the SV. Both anchors are modelled as amphipathic α-helices whereas the C-terminal region (residues 98–140, magenta) and the linker between the anchors (residues 26–64, grey) are in disordered conformations. **b** Example of TIRF images measured using 2 μM of fluorescently labelled SL-SUVs and IPM deposited onto the glass surface. Images obtained using 0 and 10 μM of αS are shown on the left and right, respectively (scale bar, 2 μm). These representative images are part of 7500 frames acquired at 40 ms of exposure time in TIRF videos. Three TIRF videos per sample condition were recorded and analyzed to quantify the number of docked vesicles and their residence time (**d**, **e** respectively). **c** Scheme of the TIRF imaging employed in this study. The focal plane extends up to 150 nm from the glass surface where the IPM are deposited (yellow membrane). SL-SUVs (pink spheres) float in the bulk solution and come into focus when docked onto the IPM surfaces. The schematic view shows measurements made in the absence (left) and presence (right) of αS and provide an example of how the double-anchor mechanism stabilises docked vesicles that are therefore imaged in the focal plane. **d** Statistical analysis of the number of docked vesicles on the IPM surface residing in the focal plane at different concentrations of αS and constant concentration of SUVs (2 μM). The symbols § and §§, indicate *p* values of 0.018 and 0.021, respectively, calculated with the unpaired *t* test using Welch's correction. Error bars report the standard deviation estimated on the triplicate measurements. **e** Residence times of docked vesicles on the IPM surface at different concentrations of αS and constant concentration of SUVs (2 μM). The symbols ** and ***, indicate *p* values of 0.033 and 0.003, respectively, calculated with the unpaired *t* test using Welch's correction. Error bars report the standard deviation estimated on the triplicate measurements.

enhancement[60,61] of the aggregation kinetics being observed depending on the experimental conditions.

Given these observations, we studied the modulation of the binding of the monomeric αS with IPM and OPM by an enrichment of the GM component (denoted as IPM-GMs and OPM-GMs, respectively). Under physiological conditions, GMs have an asymmetric distribution and partition primarily on the OPM side[62]. It is unclear, however, whether the perturbation of lipid homoeostasis observed in some neurodegenerative conditions results in an increase of GMs for both the inner and outer leaflets of the PM[52]. Our analysis showed that αS has stronger affinity to bind OPM-GMs than OPM (Fig. 3a). Higher binding strength for OPM-GMs was observed for both the N terminus ($K_D = 424.5$ μM and $L = 17.5$) and the region spanning residues 65–97 ($K_D = 3945$ μM and $L = 3.8$). CEST measurements indicated that the conformations of αS at the surface of OPM-GMs feature a primary binding site in the N-terminal region (residues 1 to 35), which binds with weaker affinity as observed in the case of IPM and SL-SUVs[43] (Fig. 3b). In addition, a secondary binding region (residues 36–98) was observed with low affinity, followed

by a C-terminal segment (residues 99–140) that lacks any interaction propensity for OPM-GMs. Taken together these data indicate that, in contexts associated with an accumulation of GMs in the membrane, including physiological lipid rafts as well as neurodegenerative conditions[52], αS changes considerably its interaction properties with OPM surfaces.

Significant alterations in the binding properties were also observed in the case of IPM-GMs, particularly in the region 65–97, resulting in nearly a sixfold increase in the interaction propensity compared to IPM ($K_D = 16.0$ μM and $L = 34.5$). By contrast, the N-terminal region of αS maintained similar strong binding for IPM-GMs ($K_D = 5.8$ μM and $L = 12.2$) as for IPM (Fig. 3c). CEST analysis confirmed that most of the alterations in the residue-specific interactions occur in the region 65–97 of αS (Fig. 3d). The other regions of αS were found to establish similar types of interactions with IPM and IPM-GMs.

We then probed whether the changes in binding propensity and conformational properties of αS at the surface of IPM-GMs affect the mechanism of stabilisation of SL-SUVs docked onto the IPM surface (Fig. 3e, f). The measurements indicated a stronger

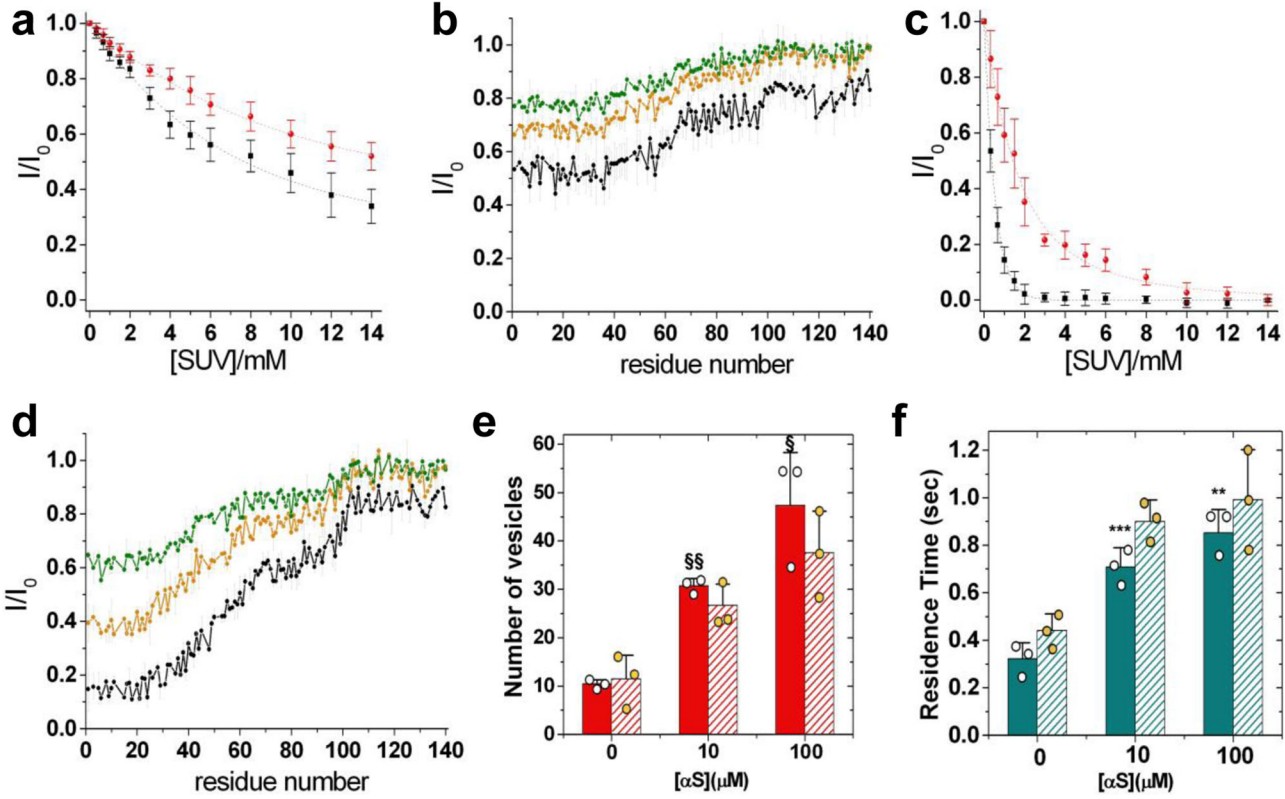

**Fig. 3 Effect of GMs on the interaction of αS with IPM and OPM. a** Binding curves of αS to OPM-GMs SUVs monitored via the signal attenuation of the peaks in the $^1$H-$^{15}$N-HSQC spectra of αS (50 μM) as a function of the concentration of SUVs. The signal attenuations have been averaged over the individual residues of the N-terminal region (black) and the region spanning residues 65–97 (red), and error bars report the standard deviation of these values. The resulting $K_D$ values are 424.5 μM ($L = 17.5$) and 3945 μM ($L = 3.8$) for the N-terminal region and the region spanning residues 65–97, respectively. **b** NMR CEST profiles along the sequence measured using a saturation bandwidth of 350 Hz (Fig. S5 for measurements performed with a bandwidth of 170 Hz) for the interaction between αS and OPM-GMs. Black, orange and green lines refer to the averaged CEST profiles measured using offsets at ±1.5, ±3.0, and ±5.0 kHz, respectively. Error bars report the standard deviation estimated on the triplicate measurements. **C** Interaction between αS and IPM-GMs (see (**a**) for details). The resulting binding affinities are 5.8 μM ($L = 12.2$) and 16.0 μM ($L = 34.5$) for the N terminus and the region 65–97, respectively. **d** NMR CEST analysis of the interaction between αS and IPM-GMs measured using a saturation bandwidth of 350 Hz (Fig. S5 for measurements performed with a bandwidth of 170 Hz). Details as in **b**. Error bars report the standard deviation estimated on the triplicate measurements. **e** Statistical analysis of the number of docked vesicles on IPM-GM surfaces (filled bars) imaged in the focal plane at different concentrations of αS and constant concentration of SUVs (2 μM). For comparison, data measured using IPM are reported with striped bars. The symbols § and §§, indicate $p$ values of 0.028 and 0.0005, respectively, calculated with the unpaired $t$ test using Welch's correction. Error bars report the standard deviation estimated on the triplicate measurements. **f** Residence times of docked vesicles on the IPM-GM surfaces (filled bars) at different concentrations of αS and constant concentration of SUVs (2 μM). For comparison, data measured using IPM are reported with striped bars. The symbols ** and ***, indicate $p$ values of 0.02 and 0.03, respectively, calculated with the unpaired $t$ test using Welch's correction. Error bars report the standard deviation estimated on the triplicate measurements.

stabilisation by αS of SL-SUVs docked onto IPM-GM surfaces, which is likely associated with an increase in the amount of αS bound to IPM-GMs than in the case of IPM. The residence times of the vesicles, however, were found to be generally shorter compared to the case of IPM, likely because of the increased electrostatic repulsion between the two negatively charged membranes. Taken together, these data indicate that alterations in the composition of the IPM can affect the modes of stabilisation of docked SVs by αS.

## Discussion

The biological activity of αS is closely linked with the conformational properties of its membrane-bound state[27]. Interactions with synaptic membranes are indeed involved in most of the putative physiological functions of αS[6,25] and are crucial for determining the toxicity of its aberrant aggregates[1,63,64]. Although the most common membrane interaction by αS occurs in the

context of SVs binding[9–14], the binding plasticity of this disordered protein enables interactions with a variety of other cell membranes, including mitochondrial membranes and the PM[26]. In particular, it has been shown that monomeric forms of αS localises intracellularly near the PM[65,66], and a number of studies have focused on the binding of the outer PM leaflet in the context of cellular uptake of αS monomers[59] and its pathological aggregates[1,67].

We showed here that the interactions of αS with the inner and outer leaflets of the PM are markedly different. The binding affinity for OPM was found to be negligible (Fig. 4a), however, upon increase of the GM content in the lipid composition, this interaction becomes considerably more favourable (Fig. 4b). This observation may be of relevance in the context of lipid rafts[54] and some neurodegenerative conditions, both associated with increased GM content in the PM[52]. A different scenario resulted from the analysis of the interaction with IPM, showing moderately high binding affinity and unique conformational properties

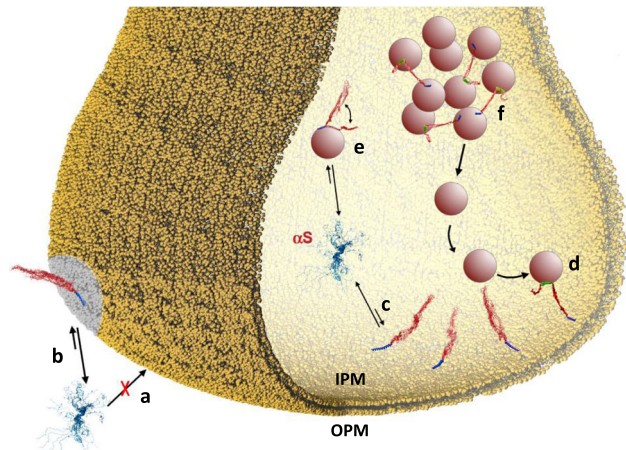

**Fig. 4 Membrane trafficking of αS at the synaptic termini.** Schematic illustration of the different interactions between αS and biological membranes at the synaptic termini. **a** αS binds to OPM with negligible affinity. **b** When the content of GMs in the membrane composition increases, such as in the case of some neurodegenerative disorders[52] and in lipid rafts[54,65], the affinity of αS for OPM is considerably enhanced. **c** The binding affinity of αS for IPM is significantly higher than that observed in the case of OPM. Upon interaction with IPM, αS adopts a conformation where only the N-terminal anchor (blue) is tightly bound to the membrane, with the region of residues 65–140 (red) having negligible association with the membrane surface. **d** This peculiar conformation has significant propensity to promote a double-anchor mechanism (N-terminal anchor in blue; second anchor spanning residues 65–97 in green) that stabilises the SV docking onto the IPM surface in an αS concentration-dependent manner. The IPM binding by αS competes with the binding to SVs (**e**), which is involved in the maintenance of pools of vesicles (**f**) from which SVs diffuse toward the active zone.

in the membrane-bound state of αS (Fig. 4c). In particular, NMR measurements revealed that αS binds IPM primarily via its N-terminal anchor, with negligible membrane interaction by most of the remainder of the protein, including the region 65–97. The weak membrane binding of this region may influence the behaviour of αS at the surface of IPM, with particular relevance for the promotion of interactions between IPM and other synaptic membranes by means of the double-anchor mechanism[17,43]. The possibility that αS could favour the docking of SVs to IPM has been discussed in literature[15,68,69], and the present study, in addition to characterising the structural basis of this process, provides direct evidence that αS stabilises both thermodynamically (number of vesicles, Fig. 2d) and kinetically (residence time, Fig. 2e) the docking of SL-SUVs with IPM in a concentration-dependent manner.

There is general debate about the promotion or inhibition of SV exocytosis by αS[15]. It has been observed that αS assists the formation of the SNARE complex via the direct interaction with synaptobrevin-2[9,10]. Other studies, however, have suggested an inhibitory effect to the SNARE activity by this protein[70–72]. At the origin of these apparently contrasting results may be the promiscuity by which αS interacts with different membranes and proteins at the synaptic termini, often resulting in competing processes. For example the stabilisation of SV docking at IPM surfaces, which we here show being favoured by αS, competes with the SV binding and promotion of SV-SV clustering by this protein (Fig. 4e, f)[14]. The competition between these interactions can be regulated by a number of factors such as local bursts of calcium ions, which favour an extended double-anchor mechanism and mediate the localisation of αS at presynaptic terminals[16], post-translational

modifications, including the phosphorylation of residues Ser 87[73], Ser 129[74] and Tyr 39[68], and modifications of the IPM composition (Fig. 3e, f).

In conclusion, this study and the available literature provide evidence for a subtle equilibrium between various interactions by αS with lipid membranes in the presynaptic space that control its biological properties. A key role in this regulation is played by the equilibrium between membrane-bound and unbound states of the region 65–97 of αS, which overlaps with the amyloidogenic NAC region. The detachment from the membrane surface of this region influences the binding affinity of αS for acidic lipid bilayers[33] as well as the efficiency by which it promotes membrane–membrane interactions via the double-anchor mechanism[17]. In addition to having a regulatory role in the biological properties of αS, conformations exposing the region of residues 65–97 from the membrane surface can also trigger its self-assembly and aggregation. This region of αS, therefore, defines a tight balance between its functional and pathological states, and our results illustrate how the impairment of this balance in PD and related synucleinopathies could be expected to be inextricably linked to age-related changes in lipid homoeostasis.

## Methods

**αS purification.** αS was expressed and purified following an established protocol[33] in which the protein was expressed in *Escherichia coli* using plasmid pT7-7[33]. In order to obtain N-terminal acetylation of αS we used coexpression with a plasmid carrying the components of the NatB complex (Addgene)[40]. After transforming in BL21 (DE3)-gold cells (Agilent Technologies, Santa Clara, USA), uniformly $^{15}$N and/or $^{13}$C labelled αS was obtained by growing the bacteria in isotope-enriched M9 minimal media containing 1 g l$^{-1}$ of $^{15}$N ammonium chloride, 2 g l$^{-1}$ of $^{13}$C-glucose (Sigma-Aldrich, St Louis, USA) and 1 g of protonated IsoGro $^{15}$N-$^{13}$C (Sigma, St. Louis, MO). The growth was obtained at 37 °C under constant shaking at 250 rpm and supplemented with 100 μg ml$^{-1}$ ampicillin to an OD600 of 0.6. The expression was induced with 1 mM isopropyl β-D-1-thiogalactopyranoside at 37 °C for 4 h, and the cells were harvested by centrifugation at 6200 *g* (Beckman Coulter, Brea, USA). The cell pellets were resuspended in lysis buffer (10 mM Tris-HCl pH 8, 1 mM EDTA and EDTA-free complete protease inhibitor cocktail tablets obtained from Roche, Basel, Switzerland) and lysed by sonication. The cell lysate was centrifuged at 22,000 *g* for 30 min to remove cell debris. In order to precipitate the heat-sensitive proteins, the supernatant was then heated for 20 min at 70 °C and centrifuged at 22,000 *g*. Subsequently streptomycin sulfate was added to the supernatant to a final concentration of 10 mg ml$^{-1}$ to stimulate DNA precipitation. The mixture was stirred for 15 min at 4 °C followed by centrifugation at 22,000 *g*. Then, ammonium sulfate was added to the supernatant to a concentration of 360 mg ml$^{-1}$ in order to precipitate the protein. The solution was stirred for 30 min at 4 °C and centrifuged again at 22,000 *g*. The resulting pellet was resuspended in 25 mM Tris-HCl, pH 7.7 and dialysed against the same buffer in order to remove salts. The dialysed solutions were then loaded onto an anion exchange column (26/10 Q sepharose high performance, GE Healthcare, Little Chalfont, UK) and eluted with a 0–1 M NaCl step gradient, and then further purified by loading onto a size exclusion column (Hiload 26/60 Superdex 75 preparation grade, GE Healthcare, Little Chalfont, UK). All the fractions containing the monomeric protein were pooled together and concentrated by using Vivaspin filter devices (Sartorius Stedim Biotech, Gottingen, Germany). The purity of the aliquots after each step was analyzed by SDS-PAGE and the protein concentration was determined from the absorbance at 275 nm using an extinction coefficient of 5600 M$^{-1}$ cm$^{-1}$. Mass spectrometry was used to confirm that the level of N-terminal acetylation was complete.

**Preparation of SUVs.** SUVs were prepared for the compositions of the inner and outer PM (IPM, OPM, IPM-GMs and OPM-GMs Table 1) using lipids purchased from Avanti Polar Lipids Inc. (Alabaster, USA) or Sigma-Aldrich (St Louis, USA) and mixed in chloroform solutions[33,35]. These include PC (Avanti code: 850457C), PE (Avanti code: 850757C), PS (Avanti code: 840034C), PI (Avanti code: 840042C), PIPs (Avanti code: 840045X), sphingomyelin (Avanti code: 860062C), cholesterol (Sigma code: C8667-1G), cerebrosides (Avanti code: 131303P), GM1 (Avanti code: 860065P) and GM3 (Avanti code: 860058P) (Table 1 for the molar fractions of the various lipid compositions)[41,42]. In addition SL-SUVs containing 31% w/w of cholesterol (Sigma code: C8667-1G) were prepared using a mixture of DOPE (Avanti code: 850725C), DOPS (Avanti code: 840035C) and DOPC (Avanti code: 850375C) at a ratio 5:3:2 as previously described[43]. In order to employ the SL-SUVs in TIRF imaging, the mixture was doped with 0.1% of 1-palmitoyl-2-(dipyrrometheneboron difluoride)undecanoyl-sn-glycero-3-phosphocholine (Top-fluor® PC). Chloroform from the lipid mixture was evaporated under a stream of nitrogen gas and then dried thoroughly under vacuum to yield a thin lipid film.

The dried thin film was then re-hydrated by adding aqueous buffer (20 mM sodium phosphate, pH 6.0) at a concentration of 10 mg ml$^{-1}$ (1.5%) and subjected to vortex mixing. In all experiments described in this paper SUVs were obtained by using several cycles of freeze-thawing and sonication until the mixture became clear[33,35]. In addition, an extrusion step was performed with a membrane with pores made of 50 nm diameter. All SUVs were controlled with DLS in order to obtain vesicles with the same average size.

**Chemical exchange saturation transfer NMR experiments**. CEST measurements[33,44–47] probed the equilibrium between membrane-unbound and membrane-bound states of αS via direct detection of saturation in the resonances of the unbound state. In studying αS-SUV interactions, CEST shows higher sensitivity than measurements based on the signal attenuation in HSQC spectra, and as a result it enables measurements at low lipid:protein ratios to minimise αS or lipid aggregation[33]. Moreover, CEST signals are directly sensitive to the interaction between αS and the membrane surface and minimise the interference from additional factors that can contribute to the transverse relaxation rates of the protein resonances[44–47]. CEST measurements were carried at 283 K to minimise the reduction of the signal-to-noise in $^1$H-$^{15}$N correlations that is generated by the water exchange of amide protons[33]. αS samples (300 μM) were incubated with the four types of SUVs (0.6 mg/ml) considered in this study (Table 1) in 20 mM sodium phosphate buffer at pH 6.0, and NMR measurements were carried out using a Bruker spectrometer operating at a $^1$H frequency of 700 MHz and equipped with triple resonance HCN cryo-probe. The CEST experiments were based on $^1$H-$^{15}$N HSQC spectra and were carried out by applying constant wave saturation in the $^{15}$N channel. Assignment of the solution NMR resonances was obtained from our previous studies[75] and controlled with a series of 3D spectra by following a published protocol[76]. Since we aimed at probing the exchange between monomeric αS (having sharp resonances) and SUVs (having significantly broader resonances), a series of large offsets was employed ($-28$, $-21$, $-14$, $-9$, $-5$, $-3$, $-1.5$, 0, 1.5, 3, 5, 9, 14, 21 and 28 kHz), resulting in CEST profiles of symmetrical shape[33,44,45]. An additional spectrum, saturated at $-100$ kHz, was recorded as a reference. The CEST experiments were measured using a data matrix consisting of 2048 ($t_2$, $^1$H) x 220 ($t_1$, $^{15}$N) complex points. NMR spectra in this investigation were acquired using Topspin 3.6.0 (Bruker, AXS GmBH, DE) and processed with NMRpipe 10.9[77] and Sparky 3.1[78].

**Transverse relaxation NMR experiments**. Standard pulse sequences were used for $T_2$ experiments[79], including the watergate sequence[80] to improve water suppression. $T_2$ values were obtained by fitting the experimental data with single exponential decays; the fitting of experimental data and the error analyses were performed with the programme Sparky 3.1[78]. Relaxation was measured at 10 °C on a sample composed of αS (300 μM) incubated with various types of SUVs (Table 1) at a concentration of 0.6 mg/ml and using a Bruker spectrometer operating at a $^1$H frequency of 700 MHz and equipped with triple resonance HCN cryo-probe. Assignment of the resonances as in the CEST measurements.

**Total internal reflection fluorescence microscopy**. TIRF microscopy can selectively excite fluorophore molecules within 150 nm from the surface of a support glass (cover slip)[81]. This technique increases the signal-to-noise and reduces the background fluorescence due to the minimisation of the excitation of fluorophores far from the cover slip[82]. This technique has been successfully applied to monitor properties of vesicles using sequential TIRF imaging[83,84]. We employed TIRF microscopy to image SL-SUVs docked onto the IPM (or IPM-GMs) surface. The experiments were made by creating an IPM (or IPM-GMs) lipid bilayer on an eight-well glass slide. The preparation of the glass slides included an incubation step of IPM or IPM-GMs SUVs overnight at 4 °C to allow the vesicles to collapse onto the glass surfaces and form the bilayer. After the overnight incubation, the glass wells were gently washed with phosphate buffer to remove the lipids in excess. The IPM (or IPM-GMs) coated glass surfaces were then incubated with SL-SUVs labelled with 0.1% of Topfluor® PC lipids. Different concentrations of SL-SUVs were tested to find the right concentration to image single vesicles using TIRF. A total SL-SUVs concentration of 2 μM was found to be optimal for single-molecule photobleaching experiments.

A custom TIRF imaging setup based on a Nikon Eclipse TE2000-U microscope (Nikon, Surrey, UK) was used. An argon ion laser (35LAP321-230, Melles Griot, USA) at 488 nm with 1.25 mW power was used for excitation. The images were acquired using an sCMOS camera (ORCA-Flash 4.0 V3 Digital CMOS camera, Hamamatsu Photonics, Japan). TIRF videos were obtained using 40 ms exposure time recordings with 4 x 4 binning and recorded continuously for 7500 frames (300 s). The last 3500 frames of the videos (140 s) were analysed to ensure that a steady state was reached after the initial step of photobleaching (160 s). All imaging experiments were performed in triplicates.

**Particle analysis in TIRF imaging**. TIRF images and videos were processed using the software ImageJ 1.52p (NIH, USA)[85] and the TrackMate plugin[86]. To analyse the images, we first subtracted the background from each frame using the rolling ball algorithm and choosing a ball radius between 1 and 2 pixels. The background subtraction also enabled to remove particles associated with SUVs from the bulk solution that are not in focus, which appear as blurry particles with lower intensity of the sharp particles associated with docked vesicles in the TIRF focal plane. Particles in focus (docked vesicles) were tracked using a spot detection algorithm. To this end, we used TrackMate with the LoG (Laplacian of Gaussian) detector and set a particle diameter of 5 pixels, with a quality threshold value ranging between 8 and 10. This procedure provided an indexing of the particles including their $(x, y)$ coordinates in each frame of the TIRF videos. $P$ value (two-tailed) statistical analysis of TIRF data was performed using GraphPad Prism 9 (GraphPad software, USA) and calculated with the unpaired $t$ test using Welch's correction.

**Residence time of SL-SUV docked onto IPM using TIRF experiments**. The residence time of SL-SUVs docked onto IPM can be quantified by calculating the ACF from sequences of vesicles images[87,88], which is defined as in Eq. 2:

$$P(\tau) = \sum_{t=0}^{T-\tau} \frac{1}{N_t} \sum_{i=1}^{N_t} \delta[w_i(t), w_i(t+\tau)] \tag{2}$$

where $\tau$ is the characteristic time of the function and $t$ is the discrete time of the image sequence. $N_t$ is the number of particles detected at time $t$ and $\delta[w_i(t), w_i(t+\tau)]$ assumes a value of 1 if the particle $i$ detected at time $t$ and at time $t + \tau$, and 0 otherwise. The ACF was fitted using a double exponential function (Eq. 3)

$$P(\tau) = a_1 e^{-\tau/t_1} + a_2 e^{-\tau/t_2} + c_0 \tag{3}$$

Where the first coefficient $t_1$ is the fast decay time constant of the ACF, which can be attributed to the stochastic loss of correlation in the fluorescence intensity of the particles, and the second coefficient $t_2$ is the slow time constant directly associated to the residence time of the vesicles on the surface of IPM.

**Dynamic light scattering (DLS)**. DLS measurements of vesicle size distributions were performed using a Zetasizer Nano ZSP instrument (Malvern Instruments, Malvern, UK) with backscatter detection at a scattering angle of 173°. The viscosity (0.8882 cP) and the refractive index (1.330) of water were used as parameters for the buffer solution, and the material properties of the analyte were set to those of the lipids (absorption coefficient of 0.001 and refractive index of 1.440). SUVs were used at a concentration of 0.05% in these measurements and the experiments were performed at 25 °C. The acquisition time for the collection of each dataset was 10 s and accumulation of the correlation curves was obtained using ten repetitions. Each measurement was repeated 10 times to estimate standard deviations and average values of the centres of the size distributions.

**Circular dichroism analysis of αS in the presence of different concentrations of SUVs**. CD measurements were made at 10 °C. CD samples were prepared in 20 mM sodium phosphate buffer at pH 6.0, by using a constant concentration of αS (10 μM) and variable concentrations of SUVs. Far-UV CD spectra were recorded on a JASCO J-810 equipped with a Peltier thermally controlled cuvette holder. Quartz cuvettes with path lengths of 1 mm were used, and CD spectra were obtained by averaging ten individual spectra recorded between 250 and 200 nm with a bandwidth of 1 nm, a data pitch of 0.2 nm, a scanning speed of 50 nm/min and a response time of 4 s. Each value of the CD signal intensity reported at 222 nm corresponds to the average of ten measurements. For each protein sample, the CD signal of the buffer used to solubilise the protein was recorded and subtracted from the CD signal of the protein. CD spectra in this investigation were acquired using Spectra Manager $^{TM}$ 2.8 (Jasco Research Ltd. CA).

**Reporting summary**. Further information on research design is available in the Nature Research Reporting Summary linked to this article.

## Data availability
Data supporting the findings of this study are available within the article and its Supplementary Information Files and from the corresponding author on request. Python processing scripts can be found at https://github.com/vrettasm/analysis_code_Man_et_al. Source data are provided with this paper.

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

## Acknowledgements

This research is supported by the European Research Council (ERC) Consolidator Grant (CoG) 'BioDisOrder' (819644, A.D.S.), the UK Medical Research Council (MR/N000676/1, A.D. and M.V. and MR/R000255/1, A.D.S.), the Centre for Misfolding Diseases of the University of Cambridge (M.V.), the St John's College Fellowship (G.F.) and Leverhulme Trust grant (RPG-2015-345, B.T. and L.Y).

## Author contributions

A.D. and G.F. designed the research. W.K.M., B.T., S.P., G.F. and A.D. performed the experiments. W.K.M., B.T., M.D.V., L.Y., A.D. and G.F. analysed the data., M.V., G.F. and A.D., wrote the manuscript. All authors revised the manuscript critically for important intellectual content and approved the final version.

## Competing interests

The authors declare no competing interests.
