## [Peer Review File · Nature Communications]

Reviewers' Comments:

Reviewer #1:

Remarks to the Author:

The manuscript "The Docking of Synaptic Vesicles on the Presynaptic Membrane Induced by α -Synuclein is Modulated by Lipid Composition" By Man et al. aims to prove a very interesting mechanism of double anchoring by alpha-synuclein that may have a great relevance for its long quested physiological function.

Few considerations are due on the manuscript:

- The cooperative behavior that emerge from the binding data is not discussed in the text. The quality of the fit is not always very satisfactory. Did the author explore the possibility of two different binding constants for the two regions of synuclein?
- A way to deconstruct the variables that contribute in this complex model (and the cooperative effects) may be to analyze the two binding region individually.
- In figure 3 the y axis of panel E and F should start at 0 and this is not a detail.
- In the discussion the authors mentioned a number of potential regulatory factors for the proposed double anchoring by alpha-synuclein, among which calcium burst. This is easy to verify in the TIRF set up together with controls with the individual domains. This functional analysis will further consolidate the proposed physiological function rather than an aspecific modification of the diffusion/interaction properties due to the presence of the protein on the surface

Reviewer #2:

Remarks to the Author:

This outstanding manuscript by Man, et al. probes the interaction of the alpha-synuclein with lipid membranes, which is critical for both its function and pathology. Specifically, the authors use NMR spectroscopy and fluorescence microscopy to study the association of alpha-synuclein with model membranes that have different compositions. While alpha-synuclein does not associate appreciably with models of the outer plasma membrane, it does associate readily with models of the inner plasma membrane. In contrast to the interaction of alpha-synuclein with synaptic-like vesicles, which involves the first ~ 90 residues of the protein, association with inner plasma membrane models occurs via only the first ~ 60 residues. This finding alone is a very interesting finding with many implications for both normal function and disease. Indeed, the authors propose that this discrepancy would provide a molecular mechanism by which alpha-synuclein could tether vesicles to the plasma membrane, wherein the N-terminal portion interacts with the plasma membrane while the central region (residues 60-90) interacts with synaptic vesicles. The authors next use TIRF microscopy to show that alpha-synuclein is able to tether synaptic vesicles to planar plasma membrane models. Finally, they demonstrate that changes in membrane composition associated with disease can alter membrane binding and vesicle tethering.

The paper appears generally thorough and rigorous, encouraging confidence. NMR data and micrographs appear high-quality and data processing appears sound. The intensity profile reported in the presence of inner plasma membrane mimics (Figure 1F) appears distinct from those reported previously by other studies (e.g., Figure S3) and does suggest release of a region previously believed to associate more fully with the membrane surface. Moreover, the model the authors propose is highly provocative and likely to be of significant interest to the field. However, before publication, the authors should address the following concerns, which are relatively minor and easily corrected:

Data presentation:

Panel 1F and 3D appear identical. The authors also do not discuss the advantages of using CEST versus earlier methods in which intensity was monitored plus/minus vesicles more simply. Figure legend 1 does not identify what the black curve is in 1F. The authors do not discuss their choice of temperature for the experiment. Would physiological T give the same result or is r.t. chosen to enhance the spectroscopy. Finally, for the general reader of Nature Commun. interested in a-syn, I am not sure the curves in D and E will be meaningful, and they could be put into the supplement. Moreover, it would be very helpful to include earlier data from SL SUV's for comparison, so the reader can easily see differences between IPM and OPM SUV's.

Major, but easily addressed in revision.

1) The authors have not accurately determined the dissociation constant for the interaction between alpha-synuclein and model membranes from their CD titrations. The authors have not accounted for the fact that each alpha-synuclein molecule interacts with multiple lipid molecules, which necessitates the use of mass action to describe the binding equilibrium (i.e., the classical Langmuir isotherm for binding of molecules to surfaces). Also, it seems that they treat the total lipid concentration as the free lipid concentration. The Hill equation requires concentrations of the free ligand or other titrant be plotted versus the fraction bound. When the $[\text{protein}] \ll K_{\text{diss}}$ one can make the approximation that the free and total ligand concentrations are the same, but that is not likely true in the current case as most reported values of K_{diss} for a-syn binding to acidic membranes are much lower than the 10 μM concentration of a-syn used in these studies. For the equilibrium under consideration it is important to use the quadratic form of the binding equation, which can be found in Galvagnion, et al. Nat. Chem. Biol. 2015, 11, 229. By this treatment one obtains both the number of lipids required to bind a protein (site size) and the K_{diss} . My expectation, however, is that the titrations are being done at relatively high protein concentration, so it will be difficult to obtain a good value of the K_{diss} . Success in fitting the CD data will also require that the titration reach saturation, which does not appear to have occurred in Figure 3A. Nevertheless, the CD spectra support the conclusions qualitatively, the true dissociation constants are likely very different.

2) It is unclear what conclusions are to be drawn from the concentration dependence in Figures 2D/E and 3E/F. The lipid concentration is not provided, so it is unclear what behavior should be expected in this concentration regime.

minor

3) Figure S3 appears to illustrate a central observation very succinctly, so it would likely be advantageous to include in the main text. Details of the data collection for the SL-SUVs (collected previously) should be provided, especially the lipid composition and protein/lipid concentrations. The caption also appears to reference the wrong publication for that dataset.

4) It would be interesting to determine if there is significance to the fine structure in the intensity profiles. If so, would it suggest something about the details of helix structure in either the bound or unbound state?

Reviewer #3:

None

Reviewer #1 (Remarks to the Author):

- a) **>The manuscript “The Docking of Synaptic Vesicles on the Presynaptic Membrane Induced by α -Synuclein is Modulated by Lipid Composition” By Man et al. aims to prove a very interesting mechanism of double anchoring by alpha-synuclein that may have a great relevance for its long quested physiological function.**

>We thank the reviewer for appreciating the significance of the double-anchor model in the context of the underlying mechanisms of function of α S.

- b) **Few considerations are due on the manuscript:
The cooperative behavior that emerge from the binding data is not discussed in the text. The quality of the fit is not always very satisfactory. Did the author explore the possibility of two different binding constants for the two regions of synuclein?
A way to deconstruct the variables that contribute in this complex model (and the cooperative effects) may be to analyze the two binding region individually.**

>We believe this point, which is also in line with the point **g** of reviewer #2, is highly relevant.

We agree that, while qualitatively capturing the differences between α S binding affinity for the inner and outer leaflets of the presynaptic membrane, the model originally employed in the fitting can be improved. To this end, we now adopted a quadratic form of the binding equation to provide both the number of lipids involved in the binding of the protein region and the K_D (as suggested by Reviewer #2). Secondly, we followed the spot-on suggestion to derive the binding constants for two independent regions of α S. To this end, we chose the N-terminal anchor and the central region 65-97, as these are two key players for the double-anchor mechanism.

As CD spectra provide averaged information across the whole protein sequence, we switched to an NMR-based approach, as this could probe the interaction at a residue specific level. In particular we monitored the intensities of the ^1H - ^{15}N -HSQC peaks of α S as a function of the membrane concentration in the sample. The new titrations have considerably improved the characterisation of the binding properties of α S and therefore we thank the referee for this comment.

- c) **In figure 3 the y axis of panel E and F should start at 0 and this is not a detail.**

>We have changed the y axis to 0.

- d) **In the discussion the authors mentioned a number of potential regulatory factors for the proposed double anchoring by alpha-synuclein, among which calcium burst. This is easy to verify in the TIRF set up together with controls with the individual domains. This functional analysis will further consolidate the proposed physiological function rather than an aspecific modification of the diffusion/interaction properties due to the presence of the protein on the surface**

> The suggested TIRF would be very interesting, however, there are limitations in performing this experiment in the presence of calcium, as it has been shown that Ca^{2+} would also trigger membrane the fusion *in vitro*, independently from the presence of proteins (Kreutzberger et al Science Advances, 2017, e1603208; Wang et al Journal of Nanomedicine 2016:11 4025–4036). As a result, fluorophore labelled lipid molecules from Synaptic-like SUVs would diffuse into the IPM bilayer, thereby altering the single vesicle imaging in the TIRF. Besides the TIRF experiment, conclusive experimental evidences already exist about the calcium modulation membrane binding properties of the C-terminal region of α S and that this has been observed in conjunction with strong localization at the

pre-synaptic membrane (Lautenschlager et al Nat Commun 9, 712 2018). Taken together these data and the present study are strongly consistent with a model of an *extended* double-anchor mechanism in the presence of calcium.

Reviewer #2 (Remarks to the Author)

a) This outstanding manuscript by Man, et al. probes the interaction of the alpha-synuclein with lipid membranes, which is critical for both its function and pathology. Specifically, the authors use NMR spectroscopy and fluorescence microscopy to study the association of alpha-synuclein with model membranes that have different compositions. While alpha-synuclein does not associate appreciably with models of the outer plasma membrane, it does associate readily with models of the inner plasma membrane. In contrast to the interaction of alpha-synuclein with synaptic-like vesicles, which involves the first ~90 residues of the protein, association with inner plasma membrane models occurs via only the first ~60 residues. This finding alone is a very interesting finding with many implications for both normal function and disease. Indeed, the authors propose that this discrepancy would provide a molecular mechanism by which alpha-synuclein could tether vesicles to the plasma membrane, wherein the N-terminal portion interacts with the plasma membrane while the central region (residues 60-90) interacts with synaptic vesicles. The authors next use TIRF microscopy to show that alpha-synuclein is able to tether synaptic vesicles to planar plasma membrane models. Finally, they demonstrate that changes in membrane composition associated with disease can alter membrane binding and vesicle tethering.

The paper appears generally thorough and rigorous, encouraging confidence. NMR data and micrographs appear high-quality and data processing appears sound. The intensity profile reported in the presence of inner plasma membrane mimics (Figure 1F) appears distinct from those reported previously by other studies (e.g., Figure S3) and does suggest release of a region previously believed to associate more fully with the membrane surface. Moreover, the model the authors propose is highly provocative and likely to be of significant interest to the field.

> We thank the reviewer for this very positive feedback.

b) However, before publication, the authors should address the following concerns, which are relatively minor and easily corrected:

Data presentation:

Panel 1F and 3D appear identical.

> While the CEST profiles shown in the original panels 1F (α S - IPM binding) and 3D (α S - IPM-GMs binding) suggest very similar binding patterns, there is an increase in the binding affinity of the region 65-97 in the case of IPM_GMs. This becomes clearer when the two panels are shown side by side.

c) The authors also do not discuss the advantages of using CEST versus earlier methods in which intensity was monitored plus/minus vesicles more simply.

> We have now discussed the advantages of NMR CEST compared to experiments probing the signal attenuation in ^1H - ^{15}N -HSQC spectra of αS as a function of SUV concentration. In particular, CEST employed in this study is based on large saturation bands applied at large offsets, which enable to probe the equilibrium between membrane-bound (NMR-invisible) and membrane-unbound (NMR-visible) states of αS . The resulting CEST signal is therefore exclusively dependent on the membrane binding and provides significant sensitivity also at low lipid/protein ratios, conditions under which protein or lipid aggregation can be minimised. The signal attenuation of the ^1H - ^{15}N -HSQC spectra, instead, is a probe of the enhanced transverse relaxation of the NMR signals. While being directly associated with αS -membrane interaction in these experiments, enhanced relaxation may additionally be triggered by other factors such as for example transient intramolecular interactions between different regions of the protein.

d) Figure legend 1 does not identify what the black curve is in 1F.

> We have amended the figure legend

e) The authors do not discuss their choice of temperature for the experiment. Would physiological T give the same result or is r.t. chosen to enhance the spectroscopy.

> We have discussed in the paper the choice of the temperature for the experiments.

f) Finally, for the general reader of Nature Commun. interested in α -syn, I am not sure the curves in D and E will be meaningful, and they could be put into the supplement. Moreover, it would be very helpful to include earlier data from SL SUV's for comparison, so the reader can easily see differences between IPM and OPM SUV's.

> We thank the reviewer for this suggestion. To improve the general readability of the paper we have now moved the curves D and E in the SI, and introduced new panels about the fitting curves from NMR titration (to answer the major point below).

g) Major, but easily addressed in revision.

1) The authors have not accurately determined the dissociation constant for the interaction between alpha-synuclein and model membranes from their CD titrations. The authors have not accounted for the fact that each alpha-synuclein molecule interacts with multiple lipid molecules, which necessitates the use of mass action to describe the binding equilibrium (i.e., the classical Langmuir isotherm for binding of molecules to surfaces). Also, it seems that they treat the total lipid concentration as the free lipid concentration. The Hill equation requires concentrations of the free ligand or other titrant be plotted versus the fraction bound. When the $[\text{protein}] \ll K_{\text{diss}}$ one can make the approximation that the free and total ligand concentrations are the same, but that is not likely true in the current case as most reported values of K_{diss} for α -syn binding to acidic membranes are much lower than the 10 μM concentration of α -syn used in these studies. For the equilibrium under consideration it is important to use the quadratic form of the binding equation, which can be found in Galvagnion, et al. Nat. Chem. Biol. 2015, 11, 229. By this treatment one obtains both the number of lipids required to bind a protein (site size) and the K_{diss} . My expectation, however, is that the titrations are being done at relatively high protein concentration, so it will be difficult to obtain a good value of the K_{diss} . Success in fitting the CD data will also require that the titration reach saturation, which does not appear to

have occurred in Figure 3A. Nevertheless, the CD spectra support the conclusions qualitatively, the true dissociation constants are likely very different.

> We thank the reviewer for this key point, which enabled us to improve considerably the quality of our data analysis. We agree that a quadratic form of the fitting provides a better model. We have therefore used the experimental conditions and equations employed in Nat. Chem. Biol. 2015, 11, 229 to probe α S-membrane interaction, which resulted in high quality fitting. We combined this key change with the suggestion by Referee #1's to derive independent binding constants for two independent regions of α S. This latter change seemed also appropriate for this study, as the membrane-affinity for IPM varies considerably in different regions of the protein.

In order to implement these changes, we switched from CD to NMR measurements, as the latter enable generating binding curves for separate protein regions by monitoring the attenuation of the intensities of the ^1H - ^{15}N -HSQC peaks of α S as a function of the IPM/OPM concentration. While this approach is less accurate than CEST measurements when studying the structural basis of the membrane interaction by α S at a residue specific resolution (see point c), when used as averaged data across whole segments of the protein sequence these measurements are robust probes to generate reliable titration curves.

2) It is unclear what conclusions are to be drawn from the concentration dependence in Figures 2D/E and 3E/F. The lipid concentration is not provided, so it is unclear what behavior should be expected in this concentration regime.

> In addition to reporting it in the main text, we have now added the lipid concentration in the panels 2D/E and 3E/F. We further discussed the conclusions associated with the observed concentration dependence.

h) minor

Figure S3 appears to illustrate a central observation very succinctly, so it would likely be advantageous to include in the main text. Details of the data collection for the SL-SUVs (collected previously) should be provided, especially the lipid composition and protein/lipid concentrations. The caption also appears to reference the wrong publication for that dataset.

> We agree that the figure S3 provides valuable information about the differences between the binding to IPM and SL-SUVs. We have moved this panel in the main text (Fig. 1) and reported the experimental conditions in the caption. We also amended the wrong numbering of the citation.

i) It would be interesting to determine if there is significance to the fine structure in the intensity profiles. If so, would it suggest something about the details of helix structure in either the bound or unbound state?

> Based on the error calculated from the measurement repeats, the fine structure of the intensity profiles in CEST should not be interpretable to obtain details of the helical-bound state, however, other studies have employed the transferred NOE to this end (Bodner et al JMB, 2009, 390:775-90). As for the unbound state, this is believed to primarily disordered, however, secondary chemical shifts in solution NMR have shown a character of transient α -helix of the N-terminal region, particularly in the acetylated form of the protein (Maltsev et al Biochemistry, 2012, 51:5004-13).

Reviewers' Comments:

Reviewer #1:

Remarks to the Author:

The authors responded to all concerns and the manuscript is now ready for publication

Reviewer #2:

Remarks to the Author:

The authors have done a splendid job of responding to the reviewers, and in the process increased the clarity and accessibility of their ms. They should be congratulated on an outstanding piece of work.

Reviewer #1 (Remarks to the Author):

The authors responded to all concerns and the manuscript is now ready for publication

> We thank the reviewer for the very positive feedback.

Reviewer #2 (Remarks to the Author)

The authors have done a splendid job of responding to the reviewers, and in the process increased the clarity and accessibility of their ms. They should be congratulated on an outstanding piece of work.

> We are thrilled with this comment and thank the reviewer for taking the time to improve our study with insightful comments.